# Nutrition and Health in Human Evolution–Past to Present

**DOI:** 10.3390/nu14173594

**Published:** 2022-08-31

**Authors:** Kurt W. Alt, Ali Al-Ahmad, Johan Peter Woelber

**Affiliations:** 1Center of Natural and Cultural Human History, Danube Private University, 3500 Krems, Austria; 2Integrative Prehistory and Archaeological Science, University of Basel, 4055 Basel, Switzerland; 3Department of Operative Dentistry and Periodontology, Faculty of Medicine, University of Freiburg, 71906 Freiburg, Germany

**Keywords:** nutrition, health, microbiome, evolution, diet, primates, hunter-gatherer, neolithization, industrial revolution, environment, behavior, cultural evolution

## Abstract

Anyone who wants to understand the biological nature of humans and their special characteristics must look far back into evolutionary history. Today’s way of life is drastically different from that of our ancestors. For almost 99% of human history, gathering and hunting have been the basis of nutrition. It was not until about 12,000 years ago that humans began domesticating plants and animals. Bioarchaeologically and biochemically, this can be traced back to our earliest roots. Modern living conditions and the quality of human life are better today than ever before. However, neither physically nor psychosocially have we made this adjustment and we are paying a high health price for it. The studies presented allow us to reconstruct food supply, lifestyles, and dietary habits: from the earliest primates, through hunter-gatherers of the Paleolithic, farming communities since the beginning of the Anthropocene, to the Industrial Age and the present. The comprehensive data pool allows extraction of all findings of medical relevance. Our recent lifestyle and diet are essentially determined by our culture rather than by our millions of years of ancestry. Culture is permanently in a dominant position compared to natural evolution. Thereby culture does not form a contrast to nature but represents its result. There is no doubt that we are biologically adapted to culture, but it is questionable how much culture humans can cope with.

## 1. Introduction

### 1.1. From the Origin of Life to the Evolution of Homo Sapiens

“Nothing in biology makes sense except in the light of evolution.”Theodosius Dobzhansky 1973

Life on Earth began with the formation of the first molecules and dates back to approximately four billion years ago. As yet, it is unclear how the primordial cell (Last Universal Common Ancestor), or LUCA, came into being. Presumably, purely chemical processes were responsible for the beginning of biological evolution [1]. The diversity of life as it presents itself today shows how successful this process has been so far. It was another milestone in the history of evolution when, about 400 million years ago (mya), vertebrates left their aquatic habitats and began to adapt to terrestrial life. The preconditions for this process included the oxygenation of the atmosphere, the ability to breathe atmospheric oxygen, alternative modes of reproduction and locomotion, access to new food sources, etc. With regard to the masticatory system, functional and constructional morphological changes had already begun in the ocean (the formation of teeth and of a secondary temporomandibular joint and increased diversification of teeth for the exploitation of specific food niches). Within the mammals, whose origins date back to 250 mya, the first primates emerged in an ecological niche 65 mya. Eight mya, the human evolutionary lineage separated from that of today’s apes. Hominins evolved from the human lineage approximately three mya with the single genus *Homo*, all species of which, except for *Homo sapiens*, are now extinct. The fossil representatives of *Homo sapiens* are now dated to 300,000 years ago [2,3].

### 1.2. Evolutionary Frameworks for Understanding Human Nature

In order to understand the biological nature of humans and their special features, one must deal with their development and look way back into the history of evolution [4]. A differentiated study of the phylogeny of our biological species allows us to better understand and assess the typical characteristics of the genus *Homo* including its habitual upright gait, the freeing of its hands, its large brain as well as its language, thinking and numerous cultural achievements. As far as humans are concerned, it is only through direct comparison with our closest relatives within a long line of ancestors that it is possible to gain a substantial insight into the evolution of *Homo sapiens* in the context of primate evolution. This limitation also explains why there is no fundamental human–animal dichotomy. Science and scientists are not immune to hubris and egocentrism. Our image of the worldviews and lives of our fossil ancestors is often shaped by our modern cultural perspective. Taking into account its biological origins, the genus *Homo* seems unique only in the sense that it implemented a revolutionary change out of the “continuum” of evolution by deliberately manipulating its ecological environment [5]. This supposedly unique step appears to have held more disadvantages than advantages for the future of the biosphere on planet Earth.

Physics can be considered the natural science, as it examines the fundamental principles that determine the processes of life in nature. Chemistry, by contrast, deals with the properties of chemical elements and compounds and with their reactions. Both disciplines together form the framework within which the biological makeup of all organisms, including humans, functions. Biochemistry represents a borderline discipline between biology/medicine and chemistry and, among other things, aims to explore metabolic processes and genetic reproduction. At the species level, two mechanisms in particular are essential: ensuring reproduction for subsequent generations (reproduction) and the physiological functional maintenance of the body in its interactions with the environment (nutrition). The chemistry of life is organised in metabolic pathways [6]. After water, carbon compounds (proteins, DNA, carbohydrates, fats, etc.) form the second most important component of a cell. The diversity of organic molecules is based on the variation of the carbon skeleton (C). Hydrogen (H), oxygen (O), nitrogen (N), sulphur (S) and phosphorus (P) frequently occur as functional groups combined with carbon; together they represent the “elements of life”.

These elements created the living conditions for the first microorganisms on Earth approximately 3.5 billion years ago. But it was not until 350 mya that a particular level of oxygen in the atmosphere had been reached and a stratospheric layer of ozone had developed which enabled higher organisms to evolve under the protection of this UV filter. The atmosphere of the Earth’s recent history was characterised by a high proportion of nitrogen and oxygen and has hardly changed since its formation. The climate was less stable, and extreme climatic crises caused by meteoritic impacts, volcanic eruptions, intense solar activity, continental drift, etc. repeatedly disturbed the balance to such an extent that several mass extinctions occurred. One such global catastrophe, which occurred 65 mya, finally paved the way for the emergence of primates, including humans, and the flora and fauna that exists today [7].

While autotrophic, photosynthetically active organisms (chemolithoautotrophic microorganisms, phototrophic microorganisms, and plants) as primary producers use sunlight as an energy source to obtain carbon from inorganic substrates such as carbon dioxide (CO_2_), heterotrophic organisms (fungi, bacteria, animals, humans) as consumers obtain the carbon required to build their own bodily substances from already synthesised organic COH compounds. Because the chemical elements that are necessary for the formation of organic matter only exist in limited quantities, life on Earth directly depends on the recycling of essential elements. These elements circulate inside organisms, ecosystems, and the biosphere through a constant cycle of build-up and breakdown processes. The driving force behind the cycle of materials is a synergy of biological, geological, and chemical processes involving the ecosystems’ biotic and abiotic components [8].

### 1.3. Nature’s Cycle of Materials and the Role of Nutrition in Sustaining Life

Living organisms are instrumental in maintaining the vital “recycling” process in biogeochemical cycles through food intake and metabolism. Through nutrient uptake, respiration, and the excretion of waste products, living organisms constantly exchange chemical components with their environment. Primary production (plant biomass) is defined as the total amount of chemical energy that is produced within ecosystems through photosynthetic activity. In terrestrial ecosystems, limiting factors include the temperature, the level of humidity and the nutrients available. Secondary production is defined as the rate at which primary consumers within an ecosystem (herbivores) convert the chemical energy of their food into their own new biomass. The importance of the trophic structure for our understanding of the dynamic processes that occur within ecosystems is emphasised by the relationship between herbivores and plants. Nutrients are moved between organic and inorganic reservoirs by means of biological and geological processes. The rate of nutrient cycling is mainly determined by the rate of decay. Nutrient cycles are strongly influenced by the vegetation. From an early phase of the Pleistocene, humans had a profound impact on the natural nutrient cycle and permanently changed the existing flora and fauna [9,10,11].

The term nutrition combines all processes that ensure the supply of substances that contain energy to a living organism. Accordingly, nutrition is a prerequisite for sustaining the life-force of every living being. Food consists of energy-rich organic compounds in solid or liquid form, which are required for the formation of cells, tissue, bones, and teeth and for maintaining the organism’s energy metabolism. Both animal- and plant-based foods contain nutrients (carbohydrates, proteins, fats) and supplements (vitamins, minerals, trace elements, fibre). Lipophilic and hydrophilic vitamins flank the metabolic functions by regulating the utilisation of nutrients. Because most of these vitamins are not synthesised by the human body, they must be ingested through food. The same applies to minerals (major minerals and trace elements), which regulate the cellular and bodily functions; they cannot be produced by the organism itself and must be supplied through nutrition. These natural inorganic nutrients occur in various chemical compounds, but the body can only ingest them from very specific ones. Major minerals (calcium [Ca], potassium [K], magnesium [Mg], sodium [Na], chlorine [Cl]) occur in the body in high concentrations, while essential trace elements (iron [Fe], chromium [Cr], cobalt [Co], fluorine [F], zinc [Zn], copper [Cu], iodine [I], manganese [Mn], selenium [Se], silicon [Si], molybdenum [Mo], vanadium [V]) occur in low concentrations. Deficiencies or overdosages of minerals result in impaired bodily functions [12]. 

From an evolutionary perspective, *Homo sapiens* is a relatively recent product of history. Its success since its emergence is illustrated, for instance, by the fact that it has managed to adapt to his environment all over the world [13]. It was supported in this endeavour by cultural evolution [14]. In addition to food and drink, the core factors that have ensured the species’ survival over the course of its history were cultural achievements such as clothing, housing, the use of energy, etc. These achievements could not have been made without cultural, social standards. And without social integration, human beings themselves would neither have been conceivable nor would they have been capable of surviving. With regard to nutrition, we cannot speak of a single, “natural” way of consuming food. It was, in fact, the very indeterminacy of the human diet, i.e., a cultural factor, that worked to the species’ advantage and was ultimately what allowed *H. sapiens* to adapt to any eco-system on Earth [15]. While the Inuit subsisted mainly on animal proteins, people in the Andes lived primarily on a plant-based diet [16]. For the majority of recent hunter-gatherers, however, well over half of their diet came from animals. Sufficient consumption of plants is advantageous for the human organism because, unlike carnivores, it cannot synthesise vitamin C on its own. However, the ascorbic acid content of fresh meat and offal is often sufficient to prevent scurvy.

## 2. Methods and Techniques for the Reconstruction of Diet in the Past

The retrospective reconstruction of our ancestors’ diet is far more difficult than it is for recent populations, offers numerous options and usually depends on a successful transdisciplinary cooperation at the interface between prehistory (archaeology), anthropology (bioarchaeology), chemistry, biochemistry, geology, and evolutionary medicine. In terms of the biohistorical source material, the starting point for the reconstruction of diet in the context of archaeological research are plant, animal, and human remains from archaeological excavations. Archaeobotany and archaeozoology study the environmental, economic, and nutritional history, the human impact on the environment and, since the beginning of domestication, the economic importance of domestic animals and cultivated plants, and also reconstruct the range of plant- and animal-based food consumed. Taking into account the chronology and the archaeological data available, diachronic conclusions can be drawn with regard to the subsistence (procurement) and consumption behaviours and insight can also be gained into social history. Reconstructions of the environment, the technical achievements, and the traces in the landscape (e.g., clearings, terracing, irrigation, stables) provide indirect evidence of the nutritional situation in different periods.

Archaeobotany is the study of plant remains, primarily from anthropogenic deposits from the past (e.g., settlements, latrines). Together with archaeozoology (see below), it is an important cornerstone in the research and reconstruction of the economic, natural and settlement history of past epochs. The source materials studied by archaeobotanists primarily include plant macroremains such as seeds, fruits, wood, leaves, stems and other parts of plants such as tubers, roots, and bulbs [17]. The most important finds categories are seeds and fruits, which can be identified to species level. In addition, plant microremains such as pollen and spores, phytoliths and starch grains are examined. In terms of the emergence of agriculture, the Neolithic period is a main chronological focus [18]. 

Animal bones are the primary source material studied by archaeozoologists. The bones mainly derive from the slaughter waste of domestic and/or wild animals consumed by humans [19]. Occasionally, additional information can be obtained from animal substances (e.g., fats, proteins) found in containers. Taken together, these sources provide quantitative and qualitative evidence of dietary trends and the availability of hunted or farmed animals. In this way, and in combination with archaeobotany and archaeozoology, it is possible to reconstruct the environment, economy, general supply, and social differences with regard to human diet in different eras [20,21]. 

Direct statements concerning human nutrition in the past are the domain of anthropology and can essentially be inferred from the preserved hard tissue remains of our ancestors such as bones and teeth. While the phylogeny of humankind (palaeoanthropology) as a discipline deal with various species and focuses on the trans-specific course of human evolution, prehistoric anthropology (bioarchaeology) investigates the intraspecific variability of the species *H. sapiens*, represented by more recent populations (from approx. 40,000 years ago in Europe). The focus of prehistoric anthropology, therefore, is not on the evolutionary process (macroevolution), but rather on recent human history, which is characterised by a gradual emergence of (supra)regional groups (archaeological cultures). The population concept of biology (microevolution) enters the discussion with the attempt to obtain evidence of the lifestyles and living conditions of past populations from their biological makeup. With that, it is no longer the individual fossil-no matter how spectacular-that is the focus of attention, but the population as a whole, which is analysed by raising questions about the general living conditions and life processes [22,23].

The human skeleton and teeth provide numerous clues about the type of food consumed and about nutritional deficiencies and malnutrition [24]. The study of prehistoric and historical diets is fundamental for the understanding of human behavioural patterns and subsistence strategies. Like animals, humans invest a considerable amount of time in procuring and stockpiling food. Then as now, the focus was on developing new strategies and techniques to achieve maximum yield with minimum effort (cost-benefit effect). Until the 1980s, fundamental information on the subsistence behaviour of past populations was derived from archaeological settlement features, technological complexes, household effects, archaeozoological and archaeobotanical evidence as well as osteological parameters (stress markers, body height, tooth wear and others). Since then, the analysis of stable isotopes found in biohistorical materials from humans and animals has opened up a new biogeochemical approach to the study of our ancestors’ diet [25]. The experimental dietary reconstruction using the stable isotopes of carbon (C) and nitrogen (N) is based on the assumption that the isotopic composition of human hard tissues can be seen as direct and constant evidence of the food consumed and that there is a measurable and systematic difference between the signals of the consumers and their food due to accumulation (fractionation), which can be detected by mass spectrometry [26] (Appendix A). To validate the isotopically determined δ13C and δ15N isotope values, the animal bones examined for comparison should always be taken from the same archaeological context.

Analyses of stable CN isotope ratios have a broad application potential within archaeology. Depending on the initial conditions and study design, they generate relevant information at an individual and collective level that provides an insight into the subsistence conditions and dietary habits of our ancestors. While the results do not allow us to reproduce detailed daily menus from the past, they do enable us to distinguish between food categories such as meat and other animal proteins versus plant-based foods, terrestrial versus aquatic sources of protein, C3 plants versus C4 plants, all of which made up our ancestors’ diet during their lifetime (Figure 1; Appendix A). No other method of analysis in anthropology can provide as much dietary information for all age groups, from the weaning of infants and the dietary changes at toddler age, to older children and adolescents to adults. In terms of the consumption of animal products, women often yield moderately lower δ15N values compared to men. Besides behavioural characteristics, numerous other factors play a role in the assessment of the nutritional balance of individuals. Apart from an individual’s social position within the community, which may be reflected in their diet, geographical and diachronic differences across groups, and even economic conditions can be identified. As well as attempting to reconstruct past menus which, however, does not allow us to distinguish between “good” and “bad” foods, the goal is to reconstruct nutritional differences within and between populations. On this basis, it is possible to make statements regarding the general state of health, the pathophysiology, nutritional stress, physical growth, and the development of diseases [27].

Dental calculus, like bones and teeth, remains preserved for millennia [29] and contains viruses and biomolecules from all areas of life [30] with the oral cavity serving as a long-term reservoir for bacteria that are responsible for local and systemic diseases [31]. Interestingly from a diagnostic point of view, systemic diseases such as diabetes are always preceded by local oral pathologies. Genetic studies on dental calculus from prehistoric and historical skeletal finds allow us to characterise specific DNA sequences [32]. This, in turn, has enabled us to identify food sources, pathogenically altered oral microbiomes, opportunistic pathogens, human-associated antibiotic resistance genes and human and bacterial proteins. Thanks to the results obtained, periodontal pathogens such as *Tannerella forsythia* have been detected genetically, which has confirmed the suspected links between host immunity factors, “red complex” pathogens and periodontal disease [30]. Usually abundant in historical burials, dental calculus thus allows us to carry out parallel examinations of pathogen and host activities on the one hand and nutritional aspects on the other. Both the identification of plant remains in dental calculus and the palaeogenetic analysis of the oral microbiome [33,34] are innovative methods for anthropologists to study the procurement of food, nutrition, the pathogenic potential, and the behavioural patterns, and have also provided evidence of medically relevant uses of certain plants [30,34]. In this respect, studies of our closest relatives are particularly interesting and informative [35].

## 3. Basics of Nutrition from Early Primates through Prehistoric Periods to the Industrial Age-from Nature-Given to Culturally Shaped

### 3.1. The Diet Spectrum of Primates

The diet of non-human primates basically includes a variety of seasonally available plant food components: fruits, nuts, barks, pith, seeds, grasses, stems, flowers, leaves, roots, and tubers [36]. Plant-based foods, which form the main part of the diet, are only occasionally supplemented by prey from within the habitat which generally includes birds and bird-eggs, insects, lizards, frogs, bats small rodents and crustacea. While the natural way of life of great apes, including their feeding habits, is relatively well known [37], there are still rather large gaps in our knowledge of other non-human primates. The diet of wild chimpanzees includes more than 80 different fruit species and more than 90 plant species; meat is rarely consumed and only in very small amounts [38]. The digestive performance of Colobinae is an outstanding example of a very specific adaptation to the food available. Native to Africa and Asia, these Old World monkeys have developed a digestive system that is unique among primates and exhibits similarities to that of ruminants. The physiological peculiarity of their digestive system allows them to consume foliage and seeds from forest and woody habitats and thus exploit food niches that are inaccessible to other mammals [39]. 

Using isotope data, researchers have been able to decipher the feeding behaviour of wild bonobos from the Congo via biochemical marker substances, allowing them to examine the composition and intake of nutrients [40]. Their diet was primarily plant-based, with clear differences only between higher- and lower-ranking males; these differences were based on social dominance and regulated individuals’ access to high-quality nutrients. Less well studied, on the other hand, are adaptations of the digestive tract to lifestyle changes, which also include the specific microflora. The red-shanked douc (*Pygathrix nemaeus*) is a primate species from the group of langurs (*Presbytini*); it is a pregastric fermenter and is difficult to keep in captivity. As part of a comparative analysis, a group of researchers from the US were able to examine the dietary components of red-shanked doucs with four different lifestyles: wild, semi-wild, semi-captive and captive, which showed that the lifestyle also alters the microbiome [41].

Carbon isotope data from early hominins show that their diet differed significantly from that of today’s great apes [42]. Four mya, hominins fed primarily on C3 resources, like modern chimpanzees. From approximately 3.5 mya, the hominins’ carbon composition began to show a spectrum that was significantly more 13C-enriched; this could be explained by an increased consumption of C4 or CAM plants (Appendix A). The diet of *Paranthropus* in East Africa even evolved towards a C4/CAM specialisation, which is otherwise unknown in catarrhine primates. Australopithecines, which are considered “pre-humans” and direct ancestors of humans, also deviated in terms of their dietary spectrum from the C3 diet typical of non-human primates, consisting mainly of fruits, leaves, and herbs. Their adaptation to a mixed and diverse C3/C4 diet seems to have occurred as a consequence of changes in their habitat brought about by climate change, which necessitated a partial dietary shift towards grasses, sedges and other plants and probably meat. Diet is considered one of the important factors in human evolution. The food consumed has an influence on the metabolic processes, the nutritional behaviours and the interactions of living beings with their environment. Henry et al. [43] was able to record the wear patterns on the teeth (dental microwear analysis) of *Australopithecus sediba* (two mya) and explore their feeding behaviours by examining phytoliths in their dental calculus.

Numerous studies have concluded that the freeing of the hands and the associated shift from a plant-based diet to one that relies more on animal protein was crucial for the development of the human brain [44]. Other studies have found that starch-containing plant foods were of equal importance for human evolution during the Pleistocene [45]. Easy to digest carbohydrates are exceedingly well suited to the increased metabolic demands of a growing brain. The invention of cooking further increased the digestibility of carbohydrates and improved their taste. When starch is cooked, more energy is made available to human tissues with high glucose requirements such as the brain, the red blood cells, and the developing foetus. Cooking is also thought to affect the salivary amylases. While starch, in its raw crystalline state, is largely ineffective, cooking significantly increases both its energy generation potential and glycaemia [31,45]. It should also be mentioned in this context that cooking increases the cariogenicity of starch [46]. 

### 3.2. Medical Significance

Caries can be observed in many species of wild non-human primates, including our closest living relatives, the chimpanzee (*Pan troglodytes*), but it is considered rare in wild populations. As part of a recent study, 11 primate taxa with *n* = 339 individuals or 7946 teeth were macroscopically examined for the presence of caries [47]. All species studied were from wild populations and came from collections in Japan and England. The overall caries intensity (Appendix B) was 3.3% (*n* = 262) of all anterior and posterior teeth, with prevalence varying between 0 to 7%, depending on the primate species. Particular attention was paid to the occurrence of caries in the anterior region because some authors excluded it as a predilection area for caries. The species with the highest rate of caries in the anterior teeth included various species of the genus langur (*Presbytis femoralis*) at 19.5%, guenons (*Cercopithecus mitis* and *Cercopithecus denti*) at 18.3%/22.4%, chimpanzees (*Pan troglodytes verus*) at 9.8% and *Gorilla gorilla* at 2.6%. The results probably reflected differences between species with regard to their diet and food processing. The processing of cariogenic fruits and seeds in the anterior area probably contributed to the high caries intensity there. However, further studies on living primate populations are required to verify this assumption.

Differences between the sexes with regard to the incidence of caries in our closest relatives, the chimpanzees, with incidences in females (9.3%) far outweighing those in males (1.8%) are extremely interesting. It is likely that the sexual dimorphism in this case reflects behavioural differences in food processing. In terms of lifestyle, the study published by Clayton et al. [41] mentioned above, detected dysbiosis in red-shanked doucs held in captivity due to an increase in microbial biomarkers such as *Bacteroides* and *Prevotella*. The study used doucs as a model species to examine the relationship between the microbial community in the gastrointestinal tract, the animal’s lifestyle, and its state of health. Faecal samples and behavioural patterns with regard to diet clearly showed a gradient in the composition of the microbiome along an axis that directly correlated with changes in the natural lifestyle. Dietary diversity within a natural environment is certainly the determining factor in microbiome diversity. As a result of captivity, the range of foods consumed was reduced, which had an impact on the composition of the microbiome.

Epidemiological comparisons of the prevalence of caries and periodontitis in great apes and in humans have shown that the prevalence in humans clearly dominates [48]. Human saliva contains a higher percentage of caries and periodontitis bacteria, a more uniform bacterial composition, and a lower bacterial diversity than that of apes [49]. Chimpanzees living in captivity underwent periodontal and cardiac examinations under anaesthesia to assess circulating markers for cardiac health and nutritional status. Although there was a significant amount of supragingival plaque on their teeth, only low levels of bleeding were observed. The neutrophils in the peripheral blood showed a similar response to innate and adaptive immune stimuli as is seen in humans. High serum levels of NT-proBNP-S were associated with subsequent death from heart disease. The dental and cardiac data did not appear to correlate [50]. 

### 3.3. The Diet of Pleistocene Hunter-Gatherers

There is no question that humans evolved from the group of primates and share a 68-million-year evolutionary history with them. Based on fossil finds from all over the world, it is also certain that all hominids older than two mya exclusively originated from Africa. There is no doubt that the upright species of the genus *Australopithecus* were the earliest ancestors of humans. What is not completely clear, however, is how and under what conditions *Homo* evolved in Africa. Presumably, the onset of changes in the world climate (cooling) 2.8 mya, the effects of which impacted Africa, had a lasting impact on the early development of the genus *Homo*. Dry savannah landscapes emerged in East Africa, which had a considerable impact on food supplies and required novel consumption habits to make the food digestible. Under these circumstances, new survival strategies and adjustments were the order of the day. Early humans with optimally equipped dentition such as *Australopithecus*, or those that had developed new behaviours such as the use of primitive tools (*H. rudolfensis*), possessed a survival advantage. The first representatives of the genus *Homo* evolved between three and two mya from the genera *A. afarensis* and *A. bahreighazali*. The debate still rages with regard to the sequence or even contemporaneity of the early fossil human forms. If *H. rudolfensis* and *H. habilis* are considered transitional hominins, the main issues revolve around *H. erectus* and *H. heidelbergensis* [51].

For the longest period of their history, humans, like all their ancestors, were wholly dependent on nature. The availability of plants and animals in the immediate vicinity and further afield and the availability of water ensured survival in the Pleistocene (beginning 2.6 mya) [52]. The diet of prehistoric humans was determined by the seasons, the availability of resources, climatic conditions, and the biotope they lived in. With a lifestyle dominated by gathering, the available food consisted primarily of plants (80%) such as leafy greens, sweet grasses, nuts, seeds, tubers, berries, roots, fruits, and pulses as well as animal proteins from wild animals and fish (20%). The plants were hard, tough, fibrous foods that were only moderately processed and therefore highly abrasive to the teeth (Alt et al. 2022, submitted). Larger brains and bodies were closely associated with the evolution of the genus Homo, which significantly increased their energy requirements. At the same time, however, the effort required to obtain the energy required decreased due to a reduction in tooth size and a shortening of the intestinal tract. This paradoxical situation with increased energy requirements on the one hand and a reduction in the masticatory and digestive capacity on the other was balanced out by an expansion of the food spectrum, in particular by the increased consumption of meat, and by the use of tools and later of fire [44]. The decrease in tooth size in *Homo* is also seen by researchers as an indication of the fact that the use of fire made food easier to chew and digest [53]. Even early Pleistocene hominids were surprisingly adaptive and adjusted their diet to regionally available resources [54]. It is possible that this flexibility facilitated the successful spread of early humans from around 2 mya onwards by setting out from their region of origin in Africa and migrating across the world [55]. The opening up of the Old World (Eurasia) was the greatest achievement of the “success story” of *H. erectus* as a pioneer species. Migration proved to be the key to the world, which is why the discovery and opening up of new habitats appears to be part of human nature.

Due to the colonisation of a variety of habitats and the adaptation to different climatic zones, it was inevitable that certain groups of our ancestors would begin to differ significantly in terms of their food spectrum, depending on the geographic region they found themselves in. A typical and classic example of this view is the Neanderthals. Despite geographically and ecologically very different environments, they are assumed to have had a dietary strategy largely based on meat. When comparing the diets of Neanderthals and early modern humans in Europe, stable isotope analyses have shown that they differed markedly in terms of their diet [56]. While Neanderthals consumed large amounts of animal proteins from large herbivores over a long period of time, regardless of the region in which they lived, modern humans in central Europe at the same time exhibited a much broader nutritional spectrum, which included both salt-water and freshwater fish [56,57]. This model was based on highly simplistic models that largely ignored the nutritional potential of plant-based diets. Recent comparative studies on the diet and behavior of Neanderthals and early modern humans from Europe, the Near East and Africa paint a different picture [58]. Plant microremains extracted from dental calculus, such as starch grains and phytoliths, indicate that all groups of Neanderthals and *H. sapiens* studied, without exception, consumed a very similar, broad range of different plant foods. The results generated suggest that the dietary ecology of Neanderthals was significantly more complex than has been previously portrayed. Thus, the statement is no longer tenable that the coexistence of Neanderthals and anatomically modern humans in central Europe from about 40,000 years ago proves that they obtained their nutrition at least partly from different food sources [56,59].

Nutrition and reproduction are basic survival strategies for all societies. The question of survival, however, is tied to a certain degree of food security, which was lacking in the Palaeolithic, the longest period in human history [60]. During the Pleistocene the survival of the human species was closely linked to the success or failure of hunting, fishing, and gathering [61]. Access to certain food resources, in turn, was dependent on environmental parameters such as climate fluctuations, which often forced humans to adjust to new living conditions. Any archaeological reconstruction of the past, however, is more easily accomplished for successful adaptations than it is for failed efforts. Due to a permanently high level of physical activity and the scarcity of resources throughout the year, food deficiencies were a constant presence in people’s lives. Prolonged food crises necessitated instant changes in the nutritional strategies and reduced female fertility. Moreover, non-sedentary lifestyles in prehistoric times also kept the number of infants to a very low level. Prior to the transition to a sedentary lifestyle and a productive economy, these factors put a limit on population growth and kept population density low.

### 3.4. Medical Significance

From a health perspective, the hunter-gatherer lifestyle of Pleistocene groups is generally considered particularly advantageous [62,63]. In physiological terms their mobile lifestyles and consumption behaviours were perfectly adapted to the human organism’s needs [64]. This prevented the occurrence of diseases that we know today as civilisation diseases, or chronic non-communicable diseases (CNCDs) [65]. Due to mobility and diet, hunter-gatherers’ musculoskeletal system was well developed, which meant that diseases of the postural and locomotor systems, including malocclusion, were rare [66]. Comparative prevalence values of infectious diseases and zoonoses even suggest that hunter-gatherer groups exhibited a much more limited spectrum of diseases of this kind than any of the later sedentary populations [67], although infectious diseases were probably the most common cause of death among hunter-gatherers.

Carious teeth and other oral diseases were among the singular findings in our Pleistocene ancestors [68,69,70] and are very rarely observed [71,72]. Isolated publications reporting high caries incidences in pre-Neolithic communities should therefore be treated with circumspection [46]. Not every cavity in a tooth can be attributed to caries. Unlike nowadays, it is not uncommon to find caries-free jaws from a prehistoric context, where one or more teeth exhibit open pulp cavities (pulpa aperta), the formation of which can be explained by excessive attrition. It makes no sense, from the point of view of dental medicine or from the point of view of palaeopathology or palaeoepidemiology, to use the term “attrition caries” to describe the phenomenon. As a part of the masticatory organ which experiences mechanical stress, the dentition is obviously subject to physiologically conditioned processes of dental wear caused by demastication, attrition and erosion. Contrary to what is often postulated in dentistry, dental hard tissue wear is not a pathological phenomenon, but rather an age-related adaptation process [73]. The ICD-10 classification of diseases is internationally recognised and should also be applied to dental anthropological research. With regard to caries, the ICD-10 Code K02.9 is a clear assessment. As regards the consumption of starchy plant foods such as acorns and pine nuts, which were pinpointed as the main culprits in the perceived high caries frequency at Taforalt, Morocco, the conclusions reached in that case are questionable [46]. Due to the appropriative economy of hunter-gatherers, starchy plant foods have always been one of the main components in their diet. The cariogenicity of starch largely depends on the degree of processing and the presence of dietary fibre. While processed starches such as those found in white flours and white rice are highly cariogenic, the cariogenicity is substantially lower in wholefood starch products such as wholegrain [74]. 

Although the oral microbiome plays a critical role in human health, little is yet known about its diversity, variation, and evolutionary history. To learn more, analyses of dental calculus samples from Neanderthals and late Pleistocene modern humans were compared to samples from recent humans and non-human primates [75]. Up to about 100,000 years ago, a core microbiome could be reconstructed that remained the same throughout the African hominid evolution and has also been observed in recent howler monkeys, suggesting that certain taxa must already have existed when the catarrhines and platyrrhines split around 40 mya. Yet, no exact host relationships are reflected in the structure of the bacterial community and the phylogeny of individual microorganisms. Humans and chimpanzees, for instance, differ considerably in terms of their taxonomic and functional compositions [76]. The microbial profiles of Neanderthals and contemporaneous modern humans, on the other hand, are similar, although they show functional adaptations in their nutrient metabolisms, including, for example, a *Homo*-specific ability to digest starch thanks to oral streptococci [34,75].

### 3.5. The Neolithic Transition and the Emergence of Civilisation Diseases

Hunting, fishing, and the gathering of plants dominated people’s lives throughout the Pleistocene. After the end of the last Ice Age and with the onset of the greatest economic upheaval in human history, the traditional way of life changed dramatically and permanently. During a warm period, and thus favourable climatic conditions, around 12,000 BC, the first permanent settlements were established in the Levant. The appropriative way of life was gradually abandoned in favour of a productive economy and a sedentary lifestyle. The transition from hunting and gathering to crop cultivation and animal husbandry led to drastic social and economic changes as well as to modifications in the material culture [77]. During the initial phase, the farming way of life remained restricted to the region of the Fertile Crescent in the Near East, effectively slowing down the spread of the Neolithic; it was ultimately achieved partly through the transfer of ideas, but much more so as a reaction to the threat posed by climatic events such as the 8.2-ka event, which marked a clearly defined, relatively short-term change in the climate [78,79]. 

The shift from a mobile, appropriative way of life to the cultivation of cereals was a slow process, initially driven by individuals, who started it by gathering wild cereal species. The entire process took place over a period of several thousand years and was highly experimental in nature. Cultivation experiments were based on wild forms of grasses and legumes such as natural lentils, from which emmer and einkorn and numerous other varieties were developed. Emmer and einkorn are ideal for arid and lean soils and require a very low level of fertilisation. Alongside the development of cultivated plants, goats and sheep were the species first domesticated; they began to increase the size and fertility of the natural pastures around the settlements. The wild forms of wheat and barley offered the advantage that they were not as easily accessible to birds as the crops with smaller grains (rye, sorghum, millet). Over the centuries, high-yield varieties of emmer and other cereals began to spread further. Optimal areas were lower-lying mountainous regions, which mitigated the hot and arid summer months and offered abundant precipitation in the winter. Secondary cultivated plants, known as “companion plants”, developed into various vegetables such as carrots, radishes, lettuce plants and herbs [17]. 

When early farmers from the Near East, having been forced to migrate by climatic events and exponential population growth, reached central Europe 7700 years ago, the new way of life quickly took hold in the new regions. The early farmers settled in the more attractive locations, began to clear the land, plant their fields, and put their livestock to pasture, initially in the surrounding woodland and later on pastureland near the settlements. Virgin landscapes were thus converted to cultivated land. While the newcomers had a certain level of know-how, they encountered different landscapes and soils, which meant that they probably experienced some setbacks and went through another lengthy phase of learning in central Europe. Based on valid palaeogenetic data, it has been shown that it was the infiltration and assimilation of foreign populations from the Carpathian Basin and later from the Eurasian steppe that resulted in the settlement of Europe by farming populations [80,81,82,83,84].

In retrospect, the Neolithic was an irreversible revolution that, besides with the apparent advantages (higher yields, relative independence from nature, a differentiated and dynamic social structure), also had numerous disadvantages (lack of genetic adaptation, limited mobility, health consequences, etc.) [85]. In addition to sedentariness, essential characteristics of the period were the division of labour, stockpiling, social change, and global dispersal. For the first time in human history, easier access to greater food supplies led to increased fertility associated with an exponential population growth [86]. The global population rose from approximately five million at the beginning of the Neolithic to almost 200 million by the turn of the eras, but it would take until the beginning of the 19th century for it to reach one billion [87]. Beyond ecological, economic, social, demographic, and epidemiological changes, the Neolithic was the starting point for later urbanisation, and a few very large settlements already existed at the time of the transition to the Holocene [88]. Large cities began to grow up in Mesopotamia, Egypt, and China from the 4th millennium BC onwards. Ultimately, however, urban growth was restricted by the level of agricultural productivity with hardly any surplus food supplies for the cities; this did not change until the onset of the industrial age. It was not just nature that culture began to take control of during the Neolithic. The period was and is the key event that helps us to understand all processes that control our physiology and reflect health aspects through lifestyle, subsistence, nutrition, and behaviour. 

With the increase in settlement density, however, new forms of human dependence on nature emerged, which became manifest not so much through weather and climatic events but through their consequences (lack of water and steppe formation, flooding, crop losses), and ultimately led to food shortages and even famine, which still occur in some parts of the world today. With the Neolithic way of life, culture replaced nature as a way of ensuring human subsistence and nutrition, and new forms of human-nature interaction developed. Rising population densities, increasing social hierarchisation, controlled access to natural resources (e.g., metal), human impact on nature and armed conflicts increased with each successive cultural epoch and economic and social pressures intensified within and between communities, especially in times of crisis. However, volatile supply situations, food crises, famine and violence are not just a spectre of the past, but since the Neolithic Revolution have become increasingly established as permanent threats [89]. 

Our knowledge about what people consumed after the introduction of farming and animal husbandry is based on evidence of crop cultivation, stockpiling and waste disposal from archaeological remains of houses, ditches, and pits. The main sources are remnants of cereals, legumes, and vegetables, either charred or from wetland preservation, as well as waste from the slaughtering of animals. As societies became more differentiated, the range of implements and tools also became more diverse and the variety of materials such as pottery, copper, bronze, and iron increased, leading to an array of sources that provide us with evidence of the production, processing, and consumption of food. The main crops cultivated by Bandkeramik farmers in central Europe were einkorn, emmer, barley, flax, lentils, peas, and opium poppies. Due to the predominance of some of the cereal varieties, which had proved to be most robust and provide the highest yields, the general variety of plants consumed declined sharply from the Neolithic onwards in favour of just a few varieties. While the appropriative lifestyle in earlier periods only allowed for the consumption of edible plants that were seasonally available, the farming way of life led to a stockpiling economy based on just a small number of food resources, which in times of crisis posed quite a risk. The reduction in food diversity is highly likely to have culminated in a reduction in the diversity of the human microbiome, which in turn can be linked to the occurrence of various diseases [41].

The consumption of carbohydrates in the central European Neolithic was mainly covered by the wheat varieties of emmer and einkorn, which were originally domesticated in the Near East, and to a lesser extent by barley. We can assume that the use of certain varieties of cereals was based on experience gathered from certain soils and climatic conditions. Hulled cereals were ground to groats using stone querns, while finely ground flour was required to make bread; bread-baking is attested to by the discovery of ovens. Over the course of the Neolithic, cereal porridge and flat breads were quickly replaced by a variety of breads and pastries. Protein-rich legumes such as lentils and peas were primarily used as dry goods, except during the harvest season; flax and poppy seeds provided fat-rich food and were therefore suitable alternatives to animal-based food [18]. The typical farm animals such as cattle, sheep, goats, and pigs were already domesticated when they arrived in central Europe and no local forms were domesticated. Animal husbandry techniques varied from one region to the next. During the Early Neolithic, i.e., the Bandkeramik Culture, cattle dominated in central Europe, while in the contemporaneous Cardial Culture of western Europe (France, Spain), goats and sheep were the main domestic animals [19], with hunting and fishing being less important. Hunting tended to increase at times of climatic deterioration to compensate for crop failures. With the rise of the feudal system in the Middle Ages, hunting became a privilege that was completely denied to the peasantry.

Numerous archaeological findings have been made with regard to soil cultivation, the range of domesticated, cultivated field and garden crops, the ways in which the plant- and animal-based foods were prepared and how milk was used [90,91,92]. This also included the use of various wild plants to supplement the cultivated crops. The extent to which wild fruits were gathered can be estimated on the basis of archaeobotanical finds [93]. The plants identified include hazelnuts, acorns, beech nuts, apples, pears, elderberries, sloes, wild rose hips, raspberries and wayfaring tree fruits. Wild fruits were used for a variety of reasons ranging from nutrient to vitamin content. Further insight has been gained from experimental archaeology. The results have shown that emmer and einkorn, for instance, were of crucial importance, even for today’s organic crop cultivation. Both cereals are resistant to various diseases and pests and also thrive on arid and poor soils. While fertilisation can increase yields, it also impairs the robustness of cereals [94]. 

Using the plant remains recovered from the Neolithic lakeside settlement of Arbon Bleiche 3 on Lake Constance in Switzerland, it was shown how the inhabitants were able to cover their calory requirements over the 14 years during which they lived there [95]. According to the model used in the analysis, cultivated plants should account for 45%, gathered plants for 30% and animal products (meat, fish, milk) for 25% of the calories consumed [96]. It was further deduced that the share of cultivated plants should cover 60% of the plant-based calorie intake while the gathered plants account for 40% [95]. The values obtained from the samples from Arbon Bleiche 3 were 63% for the cultivated plants (cereals, oil-producing crops) and 37% for the gathered plants, which almost exactly corresponds to the values expected based on the model. In order to survive the winter months, stocks of cereals, dried fruits and meat had to be safely and durably stored. In case of crop failures due to a climatic event, for example, which were quite common occurrences, the lack of cultivated plants had to be compensated by gathering wild plants and/or by consuming more meat [95].

After the introduction of stable isotope analysis of carbon and nitrogen in the late 20th century had paved the way for comprehensive nutritional studies from a methodological point of view, it became possible, in conjunction with diachronically designed bioarchaeological and palaeogenetic studies, to obtain representative results across all cultural stages [97,98]. The isotopic dietary reconstruction presented here focuses geographically on Eurasia and chronologically on the period between the beginning of the Neolithic and the Middle Ages. Once farming had established itself as the new way of life in the Neolithic, it spread fairly rapidly throughout the entire study region. While the fundamental structure of prehistoric communities is often described as egalitarian, this social equality appears to have been more mythical than real, and the same applies to nutrition. Material and social differentiation, the unequal distribution of resources and the formation of positions of power began even before the onset of the Neolithic but became more entrenched with the introduction of the sedentary, agrarian way of life and intensified even further with each new cultural change after the development of metallurgy [99]. This was due to fundamental changes in the social and economic conditions, which resulted in societies becoming more and more differentiated.

This development was not without its problems or conflicts. Early fisher-hunter-gatherer communities on the transition to the Holocene, for instance, exhibited marked differences in their burial customs and the number of individual grave goods, which seem to point to social differentiation but have not been confirmed by any other archaeological evidence or by the CN isotope data that allow us to reconstruct their diet. The discrepancy in the data is reflected by the fact that material possessions were destroyed in the context of the funerary rites, which points to a conflict between traditional egalitarian concepts and the emerging social differentiation [99]. Based on an extensive diachronic dataset (*n* = 482 ind.; 26 sites) from a period of over 4000 years from the Early to the Late Neolithic in central Europe, a slow but continuous increase in the consumption of animal proteins can be identified, which must have been linked to an increase in the consumption of dairy [100]. In the late stages of the Neolithic and over the course of the Early Bronze Age, early élites began to emerge, whose status was reflected, among other things, in the consumption of high-value proteins [101,102,103]. The emergence of advanced civilisations in the Near East and south-eastern Europe, the construction of princely seats [103] and the formation of the first cities [104] also led to differences within the strictly hierarchically structured societies, primarily due to social changes. Moreover, certain types of political societies which were based on surplus production but did not have any state like structures or ruling classes, emerged early-on [105]. Until the beginning of the modern era, society was based on upper classes that had the means to access all kinds of luxuries in terms of nutrition on the one hand, and poor, rural populations with an unbalanced low-protein diet based on cereals and vegetables on the other. In the Middle Ages, the population was still made up of serfs and free peasants, a lower and higher nobility, and an urban bourgeoisie, which consisted mainly of tradesmen and merchants, and this combination was reflected in a very complex and diverse diet. 

In principle, however, we can assume that until well into the early modern period, access to a balanced diet was largely linked to a person’s social standing. The Three Estates system (the nobility, the clergy, and the peasants and bourgeoisie) of the Middle Ages was also reflected in the dietary habits. The most significant dietary aspects that help to distinguish between the social classes are the types of meat, fish and cereals consumed, as well as the frequency and type of spices used; the latter were primarily reserved for the nobility and, with certain restrictions, the clergy. The peasants, on the other hand, mainly lived on cereal porridge or mash with lard, cabbage and bacon or pork as well as locally grown vegetables. This social order, which had emerged during the Middle Ages, remained in place relatively unchanged for many centuries and was only replaced by new patterns of social behaviour at the beginning of the industrial age [106,107].

While the beginning of the metal ages saw an increase in the range of cereals due to the addition of varieties such as oat and millet, there was also a decline in naked wheat. Beans were the most important new crop. In general, the consumption of legumes continuously increased, and this is generally attributed to population growth. In terms of livestock and meat supplies, the large Iron Age sites of Manching in Germany (1,700,000) and Basel-Gasfabrik in Switzerland (850,000) yielded large quantities of bones, which allowed archaeologists to calculate the ratio of animals present at any one time. Cattle clearly dominated over sheep/goats and pigs [108,109,110]. During the Iron Age contact with the Greek and Roman worlds led to a flourishing trade, with wine being one of the main trade commodities. Later, the Romans brought oils, spices, fish, and fish sauces to the provinces north of the Alps. While the diet of the wealthy is known from cookbooks, bills of sale and shopping lists, we know little about the diet of the rural populace, which made up 90% of the total population. The fundamental means of subsistence throughout the Middle Ages and until the advent of the potato were crop cultivation and animal husbandry as well as horticulture [111]. While the entire population were still allowed to hunt during the Early Middle Ages, this later became a privilege of the nobility. 

Besides cereal varieties such as rye and wheat for bread, which accounted for approximately 60% of the overall diet, there were special cereals such as millet, buckwheat and green spelt, which were used to make groats and porridge (c. 15% of the diet). Because of this dependence on cereals, crop failures could not be compensated by any other foodstuffs until the potato conquered Europe in the 17th century. Before the year 1000 and after 1300, climatic crises led to considerable levels of famine. These can be identified through climate proxies as stabilising or destabilising factors in cultural history [112]. Within the natural cycle, the vegetation is just as much determined by the climate as the crop yields are on the local weather. It is obvious that periods of fundamental social and political change, whether they were flanked by climate events such as the Little Ice Age or whether they are simply presumed to have occurred as in the case of the Migration Period, always had a profound impact on the population. The consequences, for instance famine and malnutrition due to crop failures, expedited the occurrence of infectious diseases, facilitated the spread of epidemics and famine, and increased mortality rates [113]. 

### 3.6. Medical Significance

There is no doubt today that the transition from an appropriative economy, which prevailed for about 99% of human evolution, to a productive economy based on farming in the Neolithic represented a step that was radical in every respect, which is why it has also been termed a revolution. Beyond the consequences of Neolithisation described above, which were undeniably positive in the medium and long term, there were also various negative effects in terms of health and mortality, which were obviously closely related to the lifestyle change [114]. In order to characterise the changes in the incidence of diseases and causes of death over time, a model was developed that captured such upheavals; they are now known as epidemiological transitions. The model describes the relationships between patterns of disease on the one hand and social, economic, ecological, and demographic conditions on the other [115]. The Holocene, or more precisely the Neolithic, marked the beginning of the first epidemiological transition. It was characterised by a sharp increase in infectious diseases compared to the Pleistocene and a relatively low level of chronic and degenerative diseases, which only slowly increased at first. Neolithic people still spent a lot of time ensuring adequate food supplies, although this required different activities to those of hunter/gatherers. Farming (clearing, ploughing, harvesting) and the sedentary lifestyle (building houses, stockpiling) were linked to increased physical activity. This subsequently led to more and more inappropriate and excessive strain on the joints, which in turn resulted in an increase in pathological changes in the form of osteoarthritis in the extremities and the spine [116].

In the past and up to the present, plagues and pandemics have had a greater impact on the world and have reduced population numbers more than armed conflicts, famine and natural disasters combined [117]. Many diseases in recent populations probably originated in the Neolithic as a consequence of the drastic change in lifestyle. Most importantly, it resulted in an increased susceptibility to pathogens, which people were less often exposed to before the onset of the Neolithic. The domestication of crops and livestock allowed people to abandon the mobile way of life and settle down. For the first time in human history, people were now in constant close contact with domesticated animals, which often even shared the same living quarters. This led to numerous zoonoses, i.e., infectious diseases that are transmitted from animals to humans and between humans by bacteria, viruses, or parasites [118]. Animals and humans together constituted an almost limitless reservoir of potential hosts. Most zoonoses originated from wild animals, and to a lesser extent from farm animals and from rodents and birds, which found new habitats in barns, houses, storage facilities and waste pits in farming settlements, spreading ectoparasites such as lice and fleas, which were difficult to control. 

The communicable diseases of these early farmers that still threaten our health today [119,120] included tuberculosis [121], plague [122], hepatitis B [123], smallpox, measles, malaria, typhus, brucellosis [124], salmonella [125] and parasites [126]. Four essential factors are mainly held responsible for the increase in infectious diseases in the Neolithic: an unbalanced diet which was low in micronutrients and rich in carbohydrates and which weakened the immune system; the close contact with domestic and farm animals, which facilitated the occurrence of zoonoses; poor hygiene and the lack of sewage systems in the settlements, which provided a breeding ground for numerous pathogens, and the increase in population density, which led to a rise in the risk of infection. Due to the extensive human impact on nature through changes to the environment, the construction of wells and cisterns, the formation of rubbish piles and the accumulation of human and animal faeces, numerous pathogens became permanent inhabitants of the settlements and caused epidemic outbreaks of typhus, yellow fever, sleeping sickness, plague, typhoid fever, cholera and endoparasitic infections.

In principle, humans are omnivores by biological design. Unlike other species, they are food generalists, which means that they do not make any specific demands on their diet. This allows them to survive under diverse conditions in almost all geographical regions by consuming a wide range of organic substances provided by the animal and plant worlds. However, this does not always make for the best conditions with regard to human health. Until a few thousand years ago, our ancestors largely covered their nutritional needs by exploiting the seasonal supply in their natural environments and were therefore perfectly adapted to this way of life. The onset of the Neolithic brought a radical change. From the point of view of nutritional physiology, the transition to an agrarian way of life, which we have now been practising for at least 480 generations, has had considerable consequences for our health, the full extent of which have only recently become apparent. As long as people lived from gathering and hunting and were highly mobile, they were physically very fit, rather tall and muscular and had a balanced diet rich in protein and vitamins from wild plants, meat, nuts and fruits, though they also had to withstand occasional periods of undernourishment. While intermittent fasting is now considered beneficial, this is a deliberate and voluntarily practice which can be abandoned at any stage [127,128]. Epigenetic evidence from a cohort study identified a transgenerational phenomenon, where, depending on the nutritional status of a participant’s father or grandfather during the slow growth period (SGP) in their youth (good versus poor food availability), the subsequent generation showed a significant increase in cardiovascular disease and diabetes [129,130]. In contrast, the diet of early farmers was initially heavily biased towards cereal consumption, which provided them with the energy required but contained little meat, fish, and vitamins, as suggested by osteological findings of stress markers and stable isotope analyses (Figure 2) [100,131].

At the beginning of the Neolithic, the consumption of animal proteins initially decreased, the variety of food plants was reduced and the proportion of starchy cereals in the diet rose sharply [100]. The changed dietary habits of the farming populations, whose diet, at least at first, was unbalanced and largely vegetarian, led to malnutrition and deficiency symptoms such as scurvy and anaemia, and weakened the immune defences [132]. The consequences of the new agrarian lifestyle occurred worldwide and affected children and adults alike [133,134,135,136]. An adverse effect of the diet, which was largely based on carbohydrates, was a rapid widespread increase in oral diseases now considered lifestyle diseases, such as caries and periodontopathies [132,137]. It stands to reason that this development was flanked by significant changes in the bacterial spectrum of the oral cavity [33,34]. Fundamental disadvantages of the revolutionary change in the diet included the occurrence of type 2 diabetes, coeliac disease and other intolerances and are only now becoming fully apparent [138,139,140]. 

Milk, today a foodstuff of global importance, has a very complex history. Animal milk has been used by humans almost since the beginning of animal domestication [30], and if milk-producing animals had never been domesticated, none of us would drink milk today. In order to digest lactose, adults need the enzyme lactase. Babies produce sufficient levels of lactase during breastfeeding. Once a child is weaned, it also loses the ability to digest lactose. With the exception of human breast milk, milk is therefore not a natural food, but a recent achievement which resulted from our cultural evolution. In a bid to overcome the indigestibility of milk in adults, some populations developed cultural practices to reduce the lactose content of milk. Other populations developed lactase persistence (LP), a genetic trait that allows continued lactose digestion after infancy. Particularly in northern European populations and in parts of Africa, a point mutation in the MCM6 gene has become established, preventing the cessation of enzyme activity after weaning and enabling adults to digest milk [141]. Basically, however, lactose intolerance is highly prevalent worldwide. Lactose tolerance in Europe decreases sharply from northern to southern Europe. In Asia and Africa, its prevalence is less than 10%. Genome-level studies have not identified a notable increase in lactase persistence in Europe until the 1st millennium BC at the earliest [142]. When lactase activity is deficient or absent, lactose enters the large intestine causing flatulence, cramps, diarrhoea and other intestinal disorders. However, these are not the only health problems caused by the consumption of milk. Epidemiological studies suggest that there is a link between fresh milk consumption and health risks that affect the birth weight and physical growth, type 2 diabetes and various tumour diseases [143].

From an evolutionary point of view, humans were well adapted to their environments prior to the Neolithic. Mobile lifestyles and natural diets met the physiological and metabolic requirements of our species in an ideal way. This prevents the occurrence of non-communicable diseases which we count among today’s diseases of civilization [65,144]. Due to their mobility, the musculoskeletal system of hunter-gatherers was well developed, diseases of the locomotor or postural systems rare. The disruption which sets in with the transition to a production economy and sedentary lifestyles in the Neolithic changes not only the diet of *H. sapiens* in a radical way, but also induces diseases like dietary metabolic disorders and intolerances [138,145,146] as well as cardiovascular diseases like atherosclerosis, heart attacks and strokes. The Neolithic marks the starting point for the occurrence of non-communicable diseases, with an ever-increasing prevalence. Today, malnutrition, excessive weight and lack of exercise are considered their causes [147,148,149]. A further side-effect of the new agricultural way of life was an increase in workloads. Due to the reduction of mobility and hard physical labour on agricultural plots since early youth, musculoskeletal robustness increased initially. On the other hand, reduced mobility and unbalanced diets produced a reduction in body height and an increase in degenerative disease [150]. Today, the mobility dilemma emerges even in infancy due to a forced restriction in movement, progresses throughout the kindergarden and school years in the form of postural defects and excessive weight and often reaches the preliminary stages of cardiovascular disease in the teenage period already. Modern non-communicable diseases are ultimately mismatch-diseases, as three million years of human evolution selection can hardly be reconciled with the predominantly culturally imprinted lifestyles of the present-day diet since the beginning of the industrial age [151].

### 3.7. The Present-Day Diet since the Beginning of the Industrial Age

The caesura in the development of nutrition which occurred with the onset of industrial food production in the second half of the 18th century was similar in its severity to the change that occurred at the beginning of the Neolithic. Both events have been characterised by terms that can be understood as process-centred and epoch-defining terms and are politically influenced: the ‘Neolithic (urban) revolution’ [152] and the ‘Industrial revolution’ [153]. The background to the Industrial revolution, which originated in England, was extremely diverse. The most important factors in the upturn were undoubtedly numerous technical innovations in the fields of precision mechanics, toolmaking, and mechanical engineering, as well as new opportunities to import various raw materials thanks to a buoyant colonial trade and corresponding markets, with equally high demand at home. Additional preconditions were the easy access to energy deposits (coal) and infrastructural aspects such as transport routes and means of transportation (railways). Another decisive factor was that the level of subsistence in western Europe was already significantly higher than in other regions of the world. This was due to certain achievements that could be viewed as part of an agricultural revolution, including the introduction of crop rotation, the expansion of fodder cultivation along with stable feeding in the winter and improvements in animal breeding, all of which led to increased yields. The relatively productive agricultural sector and the increased supply of staple foods thanks to the introduction of the potato led to a rapid increase in population numbers, which in turn resulted in an even greater demand for food supplies. Likewise, mechanisation created a surplus of labour in the countryside. Many people moved to the rapidly growing big cities in search for work in the newly established factories [154]. High crop yields, new food processing techniques (canning) and innovations in food preservation permanently changed the production of food. New preservation technologies added to stockpiling capabilities and cheap and durable food introduced a level of autonomy when it came to ensuring everyday food supplies. The disadvantages of canning large quantities of food, such as the loss of vitamins and the addition of chemical substances, were readily accepted. Together with sugar production, canned food was a prime example of a highly processed consumer good, which heralded the beginning of a new age of food consumption [155]. 

However, the cultural evolution of our consumption habits was far from over. Eating habits change along with lifestyles and under the influence of social developments. Increased globalisation and the emergence of multicultural communities led to further changes in dietary habits. In the second half of the 20th century, ready-made meals (TV dinners) first came to Europe from the US. Other trends included healthy food and fast food, which are still popular today, as are convenience food on the one hand and nouvelle cuisine on the other. With the rising numbers of migrant workers, traditional dishes from faraway countries such as India, China, Japan, Spain, Italy, the Near East, and South America conquered the central European cuisine. At the end of the 20th century, a number of food scandals and various animal diseases such as bovine spongiform encephalopathy (BSE), avian flu, swine fever, paratuberculosis caused by ruminants, etc opened up a broad public discussion about meat consumption [156] and modern pandemics [157,158]. On the other hand, the hypothesis that the consumption of meat and milk induces chronic inflammation in humans and indirectly increases the risk of colorectal and breast cancer because of Bovine Meat and Milk Factors (BMMF) found predominantly in Eurasian cattle [159,160,161] is less well known. Circular molecules of viral origin, whose genetic material is extrachromosomal present in animals, activate the DNA after the consumption of meat and milk and the transfer into human cells, and are therefore believed to be triggers for the types of cancer mentioned. Some sections of the public have reacted to factory farming, genetic engineering and the events mentioned above by turning to vegetarianism, flexitarians, veganism, and numerous other alternative diets.

In the 21st century and the era of fast-food chains and snacks, which are now the first choice particularly for younger people when it comes to their daily food intake, behaviours that used to be crucial for survival are taking a back seat. Also, the natural components are now hardly recognisable in many of our “foods”. The main components of plants that act as nutrients are their primary constituents such as carbohydrates, proteins, and fats. The secondary constituents of plants (e.g., polyphenols, isoflavones, carotenoids, glucosinolates) perform protective and defence functions. The importance of the secondary plant nutrients is subject to debate. Little is known about how they can be identified and how they work. It is generally assumed that these substances prevent diseases and are therefore necessary for human health. The focus of investigation is the question of whether the health-promoting effects of a plant-based diet are actually due to their secondary components and what role they could play as food supplements. The concept of food synergy is based on the idea that the interactions between the different food components are significant. The balance between the individual components, the metabolic processes that occur during digestion and the biological activity at the cellular level are all important. Instead of favouring food supplements, the concept involves intensifying basic nutritional research with the aim of identifying the foods that are best suited to promoting human health [162].

From a health perspective, most foods sold in supermarkets today are dispensable. Paradoxically, consumer insecurity is increasing despite the availability of an abundance and variety of foods. The clearest sign of this development is the range of diets, which have long since shattered all standards and can no longer be simply reduced to vegan, vegetarian and omnivorous. What can we still eat and what should we avoid? Does the solution lie in a return to our ancestors’ diet, as promised by the proponents of the “paleo diet”? The fact is that the increase in allergic reactions to certain foodstuffs, the rise in diet-related diseases (e.g., obesity, diabetes, coeliac disease), food scandals, the use of pharmaceuticals in livestock breeding, the dominance of industrial food and the massive increase in obesity and deficiency symptoms have unsettled the general public, as have various new diagnostic techniques which can uncover previously unknown risks. Since we are supposed to be “learning for life” from kindergarten or at least from primary school, healthy eating should be made part of the curriculum, and advertising for products that are harmful to our health should be curtailed [163].

For approximately 99% of human history, our ancestors lived with and from nature, while the period when we gradually became farmers only covered roughly 1% of human evolution, or in other words, only a blink of an eye. Our biological origins were based on tried and tested predispositions, which in our modern industrial society, however, have become risk factors, as our behaviour and lifestyles have changed drastically and are no longer in line with our biological heritage [164]. Insights from evolutionary medicine have recently improved our understanding of chronic non-communicable diseases (CNCDs) such as obesity, type 2 diabetes, cardiovascular diseases (arteriosclerosis), cancer, allergies, depression, dementia, immune diseases, and food intolerances. However, we must go beyond the proximate effects and take an in-depth look at the ultimate causes of disease and question our biological development and evolution [165]. The most important factors that trigger lifestyle diseases include inactivity or lack of exercise due to predominantly sedentary activities, as well as an unbalanced diet which suits this lifestyle and promotes malnutrition and undernourishment. 

Many of the common health problems today stem from the fact that we do not adapt our food consumption to what is most beneficial for our digestive system. We use only a fraction of the plants consumed by our ancestors and have almost reversed the original ratio between plant-based food (75%) and meat/fish (25%). The result is overnutrition, malnutrition and undernourishment [166], which is why nutrition is one of the most important risk factors in terms of health. While our earliest ancestors were more or less herbivores by nature, as indicated by their anatomy and physiology, *Homo* later evolved into an omnivorous species. In herbivores the enzymatic digestion begins through saliva in the mouth. Carnivores, on the other hand, gulp down their food, which is digested only in the stomach. In addition, fermentation chambers exist in the large intestine of both herbivores and omnivores, where the indigestible food components are broken down. The intestine of a typical carnivore also differs from that of an herbivore with regard to its length. Carnivores have a very short intestine, herbivores have a very long one, while omnivores are somewhere in between. The intestinal length in humans can be explained in terms of evolutionary history over the course of hominisation and is generally associated with an increase in meat consumption [167]. 

The interactions between the intestinal microbiome and the metabolic system, the immune system, the cognitive development, and the brain also point to indirect links with nutrition [168,169]. The gut–brain axis is a communicative system that consists of the vagus nerve, the sympathetic nervous system, and the spinal cord. Various comparative studies on indigenous groups [170] have suggested that there is a link between changes in the intestinal microbiome and physical diseases such as ulcerative colitis, Crohn’s disease, and diabetes as well as mental illnesses such as depression and anorexia nervosa. The communication between the intestinal microbiota and the brain is believed to involve the intestinal bacteria forming different groups of substances through which information is sent to the brain and which therefore have an impact on our memory, our emotions, and our ability to cope with stress [171,172]. Another study investigated the effects of a strictly western style diet (WS diet) on modern humans after animals that were fed the same diet had shown impaired hippocampal function and poorer appetite control. After just one week, the intervention group showed a hippocampus-related decline in learning and memory (HDLM) and impaired appetite control, which was strongly correlated with the HDLM decline when compared to the control group. The researchers concluded that a WS diet rapidly reduces the appetite control in humans and that this effect results in a massive increase in feelings of hunger [173].

The second basic health-risk factor is our pronounced lack of physical activity [149]. From a physiological and functional point of view, our ancestors’ evolutionary master plan matched their habits and living conditions. However, this is no longer the case today, and the effects of natural selection have also been weakened [174]. Our ancestors had to spend considerable amounts of time procuring the daily supply of food and this also required the use of energy of motion. Today, the procurement and transportation of our consumer goods no longer takes place on a daily basis and the process has been disconnected almost entirely from the requirement to move. The problem in this context is that we were originally even better programmed for motion than many animal species, but we now no longer need to move in order to survive; we do, however, need to move in order to keep well. In the past, survival was linked to physical performance and our physiology was designed accordingly [175]. 

The medical relevance in the study of our evolutionary past is not just aimed at energy sources. Vitamin C is a prime example. It supports vital bodily functions and is involved in numerous metabolic reactions in the body. As an antioxidant, it guards against cell damage and plays an essential role in the synthesis of collagen. Because the human body cannot synthesise vitamin C to a sufficient extent, it must be ingested with food. Its importance as an essential vitamin for various metabolic processes shows why most vertebrate species are capable of synthesising vitamin C. While bats and birds can reactivate their ability to synthesise vitamin C if this is needed, a few recent genera such as humans, great apes, guinea pigs and some species of fish have irreversibly lost it [176]. With regard to primate phylogeny, the ability ended with the lemurs (prosimians) around 55 mya. This was due to mutations in the L-GLO gene, which encodes the enzyme gulonolactone oxidase which is responsible for the final catalysis in the biosynthesis of vitamin C [177]. Researchers believe that the mutation is not subject to selection because the loss only affects the production of vitamin C. There is no link between loss or reactivation and the diet of the species concerned, which suggests that the loss of the ability to synthesise vitamin C is a neutral marker.

The global increase in chronic non-communicable diseases (CNCDs) is a result of the rapid and incomplete human adaptation to recent lifestyles over the last 10,000 years and is mainly related to the areas of exercise and nutrition [140]. The latter in particular is the subject of this paper. The first findings pointing to a possible mismatch between recent nutrition and human health were made as early as the 1960s [178]. The scientific breakthrough was achieved by Eaton & Konner [138], who carried out an in-depth study of our ancestors’ diet (paleo diet) from when they were hunter-gatherers. Going way beyond the core topic of nutrition, evolutionary medicine, having been established at the end of the 20th century [179], further expanded the field and has turned it into an indispensable, holistic human science [147,180,181].

### 3.8. Medical Significance

In recent decades, the number of deaths from chronic non-communicable diseases (CNCDs) worldwide has been steadily rising. According to current data provided by the WHO, approximately 41 million (72%) out of 71 million deaths are caused by non-communicable diseases every year, making them by far the most common cause of death worldwide [182]. CNCDs are characterised by the fact that they can be a burden on a person’s health for a number of years or decades, or in some cases throughout life. The idea that it is not possible to provide compelling evidence of the transmissibility of a CNCD from one person to another [183] can no longer be supported in view of recent epigenetic insight [184,185]. Epigenetic changes in this context are defined as reactions to external stimuli that lead to pathological dysfunctions and long-term changes in gene expression. While a combination of genetic, physiological, environmental and lifestyle factors are obviously at play, the role of epigenetics requires more in-depth research. 

The classification of cardiovascular diseases, chronic respiratory diseases, certain types of cancer, type 2 diabetes, dementia and others as common CNCDs or civilisation diseases is undisputed scientifically and their socio-political significance is widely recognised. However, CNCDs do not only affect older people. Many of these diseases have their roots in a person’s childhood and adolescence. In recent decades there has been a rapid increase in the number of young people with colon cancer and immune diseases. This is believed to have been caused mainly by the consumption of highly processed foods over several years [186,187]. Until recently, oral pathologies such as dental caries, periodontal disease, certain oral cancers, leukoplakia, and other dental diseases were not counted as CNCDs [188]. This has changed and we now have a different understanding of the prevention and treatment of oral diseases [189], due, on the one hand, to new molecular insights into the oral microbiome [190] and a change in how we view the oral biofilm [191] and, on the other, to the fact that confirmed links have been observed between systemic and dental CNCDs [189,192]. These observations have led to various new hypotheses concerning the aetiology and prevention of these diseases, which is why the WHO and the FDI now consider oral health to be a key indicator of overall health [183,184,185,186,187,188,189,190,191,192,193,194,195].

In terms of the nutritional mismatch discussed above, numerous clinical studies carried out in recent decades have gone far beyond the ongoing media discussion to highlight the importance and significance of various different diets (paleo diet, Mediterranean diet, etc.) with regard to CNCDs and public health, and we now have a broad body of relevant data [66,196,197,198]. Results obtained from bioarchaeological, palaeomedical [100,199], palaeogenetic [33,200] and ethnographic [170,201] studies were particularly well received within the fields of oral medicine and dentistry. The medical aspects of human evolution and the study of the ultimate causes of certain diseases are currently a particular focus of evolutionary medicine, and it must be emphasised that evolutionary medicine, which for a long time was no more than a theoretical concept, has now gone far beyond the stage of purely providing explanatory models [181,202,203].

Promoted by the WHO [194] and the FDI [195] at a socio-political level and in response to a global demand from a health policy perspective, two central tasks have been set to be achieved by 2030: the establishment of innovative guidelines in the field of (dental) medicine and the development of new approaches to preventive healthcare. Evolutionary medicine will play a part in implementing the initiatives. The measures adopted are very welcome, even though many research questions in evolutionary medicine have yet to be answered. There is now a broad consensus, however, that nutrition was an essential factor, if not even the key factor in the emergence of lifestyle or civilisation diseases. It also seems certain that the emergence of civilisation diseases (CNCDs) was a direct result of human evolution and the modern lifestyle, that they have an impact on biological systems and can be viewed as deficits or incomplete adaptations on the part numerous organ systems. Recent studies on coronavirus, for example, have shown that the most common comorbidities of a COVID-19 infection are chronic heart disease, diabetes mellitus and non-asthmatic chronic lung disease, all of which are serious conditions that can be mitigated or prevented by a healthy lifestyle, which is why prevention is crucial [204].

From this point of view health and disease must be re-evaluated. Fundamental facts must be questioned, including the fact that pathogenic genes and mechanisms like uninhibited cell growth were never eliminated by natural selection over the course of evolution. What might be the evolutionary benefit or advantage of the existence of cancer, inflammatory processes or immune diseases for individuals or populations? Our current way of life is increasingly determined by culture rather than by our origins millions of years ago (nature) (Table 1). 

Compared to natural evolution, cultural evolution is forever in the fast lane. Evolution does not strive for perfection, nor does it give priority to quality of life and longevity. Culture is not the opposite of nature; it is a result of our nature [205]. Without wanting to sound overly pessimistic with regard to culture, we are undoubtedly adapted to culture by virtue of our biology [206]. What is unclear, however, is how much culture humankind can cope with.

### 3.9. Nutrition as a Social Phenomenon

Along with reproduction, nutrition ensures human survival and represents a basic need [207]. Eating and drinking are a fundamental part of communal life and have always been closely linked to human cultural evolution [208]. For the majority of our history, the daily struggle for food has determined people’s lives. In some sections of the world’s population this is still the case today, and millions of people continue suffer from hunger and medical conditions that are caused by malnutrition. In the western world, on the other hand, there is an oversupply of food. Never before has there been such a broad selection of foodstuffs. Yet the majority of people in the West resort to convenience food and prefer to avail of the supply of readily available fast food. With this type of diet, the name says it all, and consumers must naturally pay the price [209]. Nevertheless, we happily accept the detrimental effects on our health from the consumption of cheap, highly processed food if it means that we will have more money for leisure pursuits. Only a small section of the population will adapt their eating habits to achieve a healthy, varied, and balanced diet that suits their lifestyle. The endless range of food concepts and trends, however, can at times sound like religious creeds and have become part of a mythical world of healthy living.

The long-term consequences for our health and morbidity rates of the unhealthy diet, which is widespread in the West despite the boom in organic produce, cannot be determined, nor can we say how quickly the situation will evolve. Theories and models describing the epidemiological transition [210] attempt to record the changes in the prevalence of diseases and causes of death and consider them to both a consequence and a cause of the demographic transition. However, the rapid course of cultural evolution in recent times has forced us to continuously adapt the epidemiological transition model to ever-changing circumstances [211,212]. The importance of studying our origins and our past is clearly demonstrated by evolutionary medicine [181,213]. It is not yet possible to identify any real trends in the field of medical science, but new approaches to the prevention and treatment of these conditions are beginning to take hold. Nutrition seems to play a key role in this, and the sector of oral medicine is at the centre [214,215,216]. However, we have not quite reached the goal that Thomas A. Edison (1847–1931) is said to have foretold at the beginning of the 20th century: “The doctor of the future will give no medicine but will instruct his patient in the care of the human frame, in diet and in the cause and prevention of disease.”

## 4. Conclusions

Our recent way of life, as far as nutrition is concerned, originated in the beginning of the Anthropocene (Holocene) about 14,000 years ago with the domestication of plants and animals. During the period preceding this revolutionary transition, which accounts for 99 percent of human history, subsistence and nutrition were completely different. The basis of nutrition for non-human primates and prehistoric humans alike was the gathering of wild plants and hunting terrestrial and aquatic animal species. Thus, human beginnings are characterized by an exclusive exploitation of wild dietary resources, similar to that of wild primates. By increasing their cognitive and technical abilities, humans then tapped into further food sources, enabling production but permanently changing the environment. 

According to data updated in 2021 by the US Population Reference Bureau, based on Haub [217], an estimated 117 billion people populated the earth to date. Of these, about 60% lived as hunter-gatherers, about 35% practiced agriculture, and only a few percent have been members of industrial societies. Before plant cultivation and animal husbandry fundamentally changed the food situation, the world’s population at that time-extrapolated to be about 4 million people-subsisted solely on hunting and gathering. By 1500 AD, the end of the Middle Ages, hunter-gatherer habitats had already shrunk considerably in favor of ever-expanding farming societies. Hunter-gatherers then made up only about 1% of the world’s total population of about 425 million people at that time. Today, the hunter-gatherer way of life is nearing extinction [218].

Nutrition ensures the basis of metabolism and serves to generate energy and maintain vital functions. Although this important connection was not known to prehistoric humans, survival or self-preservation is achieved by satisfying the urge to eat. This begins in mammals after birth and consists in the instinctive sucking of their mother’s milk. The use of further food sources is enormously important and is acquired via learning effects. Changes of habitat due to changing environmental conditions involves adaptation to new foodstuffs and animals must what is edible and what is not. *H. sapiens* has become increasingly biologically and culturally detached from nature and, in the industrialized world, has to spend little time organizing daily meals. Little more than 250 years ago, 95% of humanity lived in the countryside, with most people providing for themselves from their gardens, fields and stables and where the only way to buy or barter food for sustenance was at markets. With the end of the Middle Ages, trade relations intensified, but only satisfied the needs of a small upper class. Industrialization gradually put an end to the agrarian life for the majority of the population. The populations were increasingly concentrated in the industrial centres and the countryside became a supplier of foodstuffs. New ways of food preservation as well as an increasing food processing becam widespread. These cultural developments have moved us further and further away from our natural diet. 

The preferred sources of energy for our omnivorous ancestors were primarily plant foods, complemented by fat and proteins from animals. In order to sustain life in the long term, essential nutrients such as vitamins were also necessary in the diet. The energy requirements of humans vary according to strain, and it is likely that they were at a high level for our ancestors. However, due to lack of food security, people may have suffered from deficiencies for short or long periods of time. What was normal was not so much abundance as recurring scarcity. Because of food insecurity, our ancestors therefore often met their energy needs above the minimum so as not to endanger their existence. The first clear nutritional change in mankind begins in the Neolithic. In its beginnings, the farming way of life can be seen as a crisis [219]. With the introduction of the production economy, the size of social groups increased as well as population density. More reliable food sources and settled communities were the cause of this. Economically, productivity increased, and stockpiling became more predictable. Qualitatively, however, the change in diet led to nutrient deficiencies, as previously broadly mixed diets were abandoned in favor of a one-sided diet reduced to a few starch-rich plants. 

The present food situation-both in the industrialized nations and globally-is not comparable with the past. Demographic aspects are primarily responsible for this. World population and world nutrition are directly interdependent in a complex way. More people automatically mean a higher demand for food and water. More high-quality food means more factory farming, overexploitation of land and water, decrease in biodiversity and destruction of natural habitats. In agrarian societies from ancient to early modern times, mortality was high, flanked by famine, war, and epidemics. Population growth and decline alternated. In central Europe, the Little Ice Age and the plague reduced the population [113]; in the Americas, epidemics within indigenous populations depopulated vast areas [220]. With industrialization, the world population grew disproportionately. Even underestimated rate, the world population roughly tripled between 1804 and 1960 [221]. Pessimists and optimists compete in the discourse about sustainable “carrying capacity and food security” on earth [222]. Cultivation of new agricultural land and more intensive use of it is currently counteracted by climate change. A reversal of population trends is not in sight, and the world’s population will continue to grow, only slightly slower according to current projections. More people not only increase population density, but also increase the demand for food, water, and medical care, as well as for jobs and prosperity.

Perhaps the most significant aspect of the present concerns the relationship between nutrition and health. Increasingly, more and more people around the world are suffering from various diseases of civilization, from diet-related intolerances, and are suffering from malnutrition despite the wide range of food available. At least in Western countries, there are fewer and fewer people in recent decades who must perform strenuous physical labor, but they still eat as if they worked in heavy industry. In addition, the calories supplied from processed foods have a high energy density but provide hardly any fiber and micronutrients. The failure to adapt diet and caloric intake to predominantly sedentary lifestyles and lack of exercise has consequences, the roots of which today often lie in childhood. Nutrition today is about its quality. Food should predictively promote health and help avoid diseases. This is why countless food trends and concepts are promoted, such as the veggie boom, the paleo diet, volumetrics, the Mediterranean diet and superfoods. Healthy living means consistently decimating or eliminating risk factors. But far too often we end up eating highly processed fast-food dishes and are addicted to sugar. However, the medical effects of our current diet in the form of CNCDs are much more extensive than the most common diseases such as diabetes, cardiovascular disease, and stroke, as well as caries, periodontal disease, and others. For example, the consumption of predominantly soft, highly processed food has made the teeth’s job of grinding food almost obsolete, with fatal consequences that have not really been recognized. Since we no longer abrade our teeth, like we do all other body tissues, they may be causing diseases whose causes remain unrecognized because we lack a view into the past [223]. Bruxism is such a modern disease, which causes high costs in the health care system and is probably related to our changed consumer behavior, but whose importance modern dentistry fail to recognize.

Everything in nature is in a constant state of change, including the food sector, but this change is taking place much more slowly than the accompanying cultural evolution in humans. Our origins lie in the distant past, but a look back into phylogenesis and the stages of human development is worthwhile. Optionally, evolutionary medicine provides a forum to look more closely at our biological origins and development [224]. A bio-evolutionary point of view offers new perspectives on diseases, some of whose origins can be traced back millions of years. Health and disease are equally a legacy of this evolution. Symptom-based diagnosis and therapy, as well as occasional questions about the immediate (proximate) causes of disease, characterize everyday medical practice. Questions about the real (ultimate) causes of diseases and about their prevention often take a back seat. Terms such as “diseases of civilization” and “lifestyle diseases” characterize contemporary phenomena whose occurrence is seen as a consequence of our current lifestyle. They spread faster than medical science develops. This is mainly due to the fact that today’s food is no longer based on the natural availability of food items, but on the powerful interests of food industries. Ultra-processed food means pleasure gain and further increased consumption. Originally created as an accidental product of nature, humans have increasingly withdrawn from the natural cycle by means of their culture, striving for self-sufficiency rather than modesty and sustainability. Since the emergence of humans in Africa, there is a certain inevitability in this development and there is a lack of humility to adapt cultural development to the rhythm of nature, probably with ultimately fatal consequences.

## Figures and Tables

**Figure 1 nutrients-14-03594-f001:**
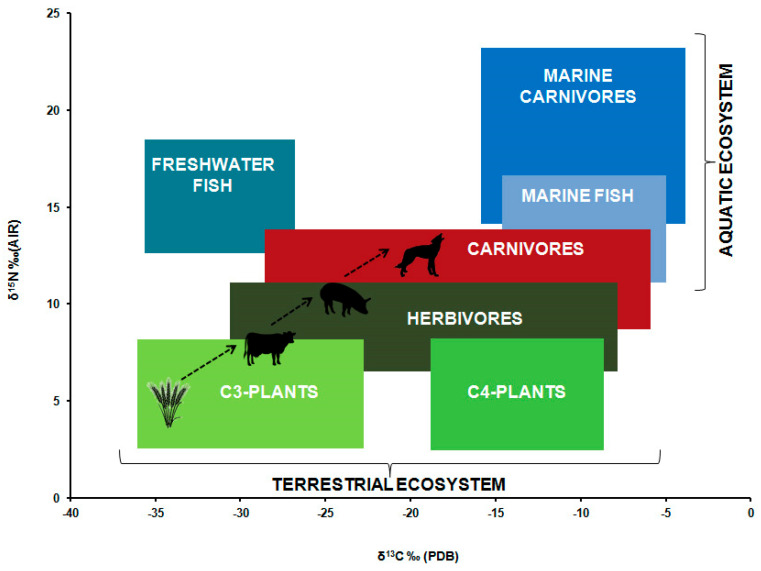
Terrestrial and aquatic food-web model for terrestrial and aquatic ecosystems with overlap ranges of carbon and nitrogen isotope values of different producers and consumers. According to the ecological niche model, food webs in different ecosystems are controlled by natural laws and work on the principle that each food level (e.g., plants, herbivores, omnivores, carnivores in the terrestrial ecosystem) occupies a particular niche in the overall food chain as exemplarily illustrated here (see also Appendix A). The trophic-level effect in the model is reflected by the accumulation of nitrogen in the food chain when individuals consume plants or animal products. The δ15N in human bone collagen is accumulated by an average of about three per mill compared to the fauna consumed. However, the complete nutritional spectrum of individuals and populations is based exclusively on a combination of the δ13C and δ15N values (mod. after Schoeninger & De Niro [28]).

**Figure 2 nutrients-14-03594-f002:**
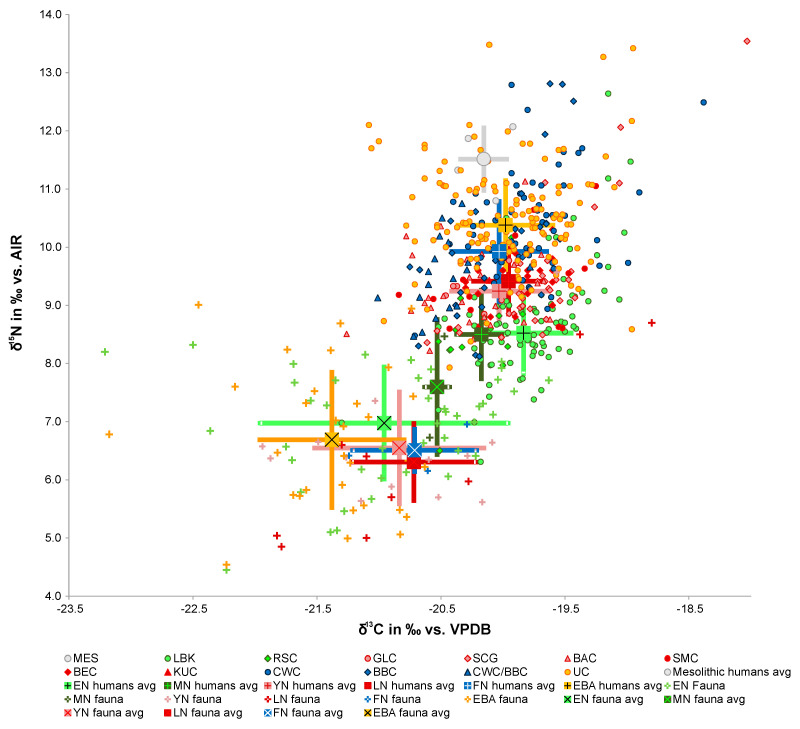
Stable isotope values of bone collagen samples from humans (*n* = 482) and animals (*n* = 109) from 26 Neolithic sites of different cultural groups in central Germany as well as one Mesolithic sample from Bottendorf, Thuringia (MES); Early Neolithic = light green, Middle Neolithic = dark green, Younger Neolithic = light red, Late Neolithic = dark red, Final Neolithic = blue, Early Bronze Age = orange. Each graph point represents one individual, either human or animal. The central finding is an increase in animal protein consumption (d15N) over time (mod. after Münster et al. [100]).

**Table 1 nutrients-14-03594-t001:** Chronological survey of food acquisition and dietary behaviour in nonhuman primates and human groups from prehistory and the Neolithic transition to a farming lifestyle up to the Industrial Revolution and the present.

	Wild Primates	Pleistocene Hunter-Gatherers	Neolithic Period	Bronze Age/Middle Ages	Post Industrial Revolution
**Way of Life**	mobile hordes	nomadic; small egalitarian groups	sedentary; agricultural groups; profound social and cultural change	sedentary; agricultural societies; increasing social differentiation and first elites; rise in violence	sedentary; industrial societies; social stratification; large disparities in wealth
**Economy**	exploitation of wild resources	systematic exploitation of wild resources including aquatic foods	production based economies; crop cultivation and animal husbandry; decreasing role of wild foods	production based economies; metalworking; advancement of farming and animal husbandry; hunting as privilege of nobility	global economy with marked interdependencies; genetic engineering of foodstuffs; diets; strong income-dependency of food choices
**Dietary** **Description**	variety of seasonally available plant food supplemented by small animals	variety of seasonally available plant food supplemented by hunting and fishing; low processed foods; occasional periods of famine	intense consumption of cereals supplemented by vegetables and domestic animals; low proportion of meat; few wild animals; few dairy products; low processed food	cereal species diversification; extension of horticultural crops; more meat consumption; more dairy; mainly low processed food	global diets; cheap meat from factory farming is popular; bread from white flour is staple food; primarily highly processed food; healthy foodstuffs are costly; organic farming is expanding; diverse food fads
**Food Preparation**	none	processing with stone and bone tools; fire use; fermentation of vegetable and animal foodstuffs	fireplaces for cooking and baking; ceramic cooking vessels	ovens for cooking and baking; metal items for food preparation and consumption	increasingly industrialized cooking; choice between grandma’s kitchen and molecular cuisine
**Medical Significance**	ideally biologically adapted; use of medicinal plants; very low rates of caries	ideally biologically adapted; low birth rate; low population density; communicable diseases low; chronic non-communicable disease absent; very low rates of caries	high proportion of starchy foods; high birth rate; population density increases; close contact between humans and animals; increase in communicable and degenerative diseases; first civilization diseases; increase in oral diseases	high proportion of starchy foods; high birth rate and population density; rise in infectious diseases; epidemics; continued increase in civilization diseases; further increase in oral diseases	population overshoot; chronic non-communicable diseases as main cause of premature death; global pandemics; increase in mal- and undernutrition; high rates of caries and periodontal diseases

## Data Availability

Not applicable.

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
