# Peer review of "Nutrition and Health in Human Evolution–Past to Present"

_nutrients, 2022, doi:10.3390/nu14173594_

Round 1

Reviewer 1 Report

1. In review article the use of titles like Material and Methods and results looks inappropriate as it is not a research article. my suggestion change these titles/heading with appropriate headings 

2. Figure 1 is less explanatory which need more labels for better understanding and caption is too long to got its understanding. improve this figure

3. Review is more subjective than summarized in the form of table information. It requires some table information on prehistoric evolution and results 

4. A meaningful summary/conclusion is needed to conclude the review. 

Author Response

  1. In review article the use of titles like Material and Methods and results looks inappropriate as it is not a research article. my suggestion change these titles/heading with appropriate headings 

Thank you for this comment. We have changed the chapter titles.

  1. Figure 1 is less explanatory which need more labels for better understanding and caption is too long to got its understanding. improve this figure

Ok, we understand the problem, but the legend is bad to cut. We put a reference to Appendix A in the text and changed the legend slightly. With the reference to Appendix A it should be understandable now.

  1. Review is more subjective than summarized in the form of table information. It requires some table information on prehistoric evolution and results 

Thank you, we have taken the hint and created a detailed table that is chronological.

  1. A meaningful summary/conclusion is needed to conclude the review.

We have also implemented this advice and added a Conclusions chapter under 4.

Reviewer 2 Report

The review article by Alt et al., has elaborated on the role of nutrition on the health aspects in different phases of human evolution in an interesting way. The article is well written and scientifically describes different aspects of nutrition and their role in health throughout human evolution, and I believe, this article will be an interesting contribution to the field. 

This article can be accepted with the following very minor modifications:

1. Please add ..400 million years ago (mya) at line 37

2. Please check all the genus and species names and represent it in "Italics".

3. The authors can add a paragraph on the start and progression of chronic non-communicable diseases in the "(Oral) medical significance" section under the "The Neolithic transition and the emergence of civilization diseases" as an increase in sedentary lifestyle and change in food habits during the Neolithic revolution has a greater impact on several NCDs which is still continuing. So this could be of particular interest to many readers.

4. "....though they also hat to withstand.." Please check hat or had in line no: 787.

Author Response

Thank you for your helpful comments.

  1. Please add ..400 million years ago (mya) at line 37

Note was settled

  1. Please check all the genus and species names and represent it in "Italics".

Unfortunately, the program probably did not accept our species names when we entered them. This has now been corrected.

  1. The authors can add a paragraph on the start and progression of chronic non-communicable diseases in the "(Oral) medical significance" section under the "The Neolithic transition and the emergence of civilization diseases" as an increase in sedentary lifestyle and change in food habits during the Neolithic revolution has a greater impact on several NCDs which is still continuing. So this could be of particular interest to many readers.

Thank you for this suggestion, which we were very happy to implement.

  1. "....though they also hat to withstand.." Please check hat or had in line no: 787.

Has been done.